# Contrastive Mixture of Posteriors for Counterfactual Inference, Data Integration and Fairness

## Abstract

Learning meaningful representations of data that can address challenges such as batch effect correction, data integration and counterfactual inference is a central problem in many domains including computational biology. Adopting a Conditional VAE framework, we identify the mathematical principle that unites these challenges: learning a representation that is marginally independent of a condition variable. We therefore propose the Contrastive Mixture of Posteriors (CoMP) method that uses a novel misalignment penalty to enforce this independence. This penalty is defined in terms of mixtures of the variational posteriors themselves, unlike prior work which uses external discrepancy measures such as MMD to ensure independence in latent space. We show that CoMP has attractive theoretical properties compared to previous approaches, especially when there is complex global structure in latent space. We further demonstrate state of the art performance on a number of real-world problems, including the challenging tasks of aligning human tumour samples with cancer cell-lines and performing counterfactual inference on single-cell RNA sequencing data. Incidentally, we find parallels with the fair representation learning literature, and demonstrate CoMP has competitive performance in learning fair yet expressive latent representations.

## 1 Introduction

Large scale datasets describing the molecular properties of cells, tissues and organs in a state of health and disease are commonplace in computational biology. Referred to collectively as 'omics data, thousands of features are measured per sample and, as single-cell methodologies have developed, it is now typical to measure such features across $10^5$–$10^6$ samples [1, 2]. Given these two properties of 'omics data, the need for scalable algorithms to learn meaningful low-dimensional representations that capture the variability of the data has grown. As such, Variational Autoencoders (VAEs) [3, 4] have become an important tool for solving a range of modelling problems in the biological sciences [5, 6, 7, 8, 9, 10]. One such problem is utilising representations for counterfactual inference, e.g. predicting how a certain cell or cell-type, observed only in the control, would have behaved when exposed to a drug [9, 10, 11]. Another key problem is removing batch effects—spurious shifts in observations due to differing experimental conditions—from data in order to integrate or compare multiple datasets [5, 12, 13, 14, 15].

We present a formal account of these challenges and show that, to a great extent, they can be seen as different aspects of a the same underlying problem, namely, that of learning a representation that is marginally independent of a condition variable (e.g. experimental batch, stimulated vs. control). Figure 1 [CoMP] illustrates what this looks like in practice: the complete overlap of the cell populations from different conditions in the latent space. This directly addresses batch correction, and in the case where we also have a generative model that maps from latent space back to the original

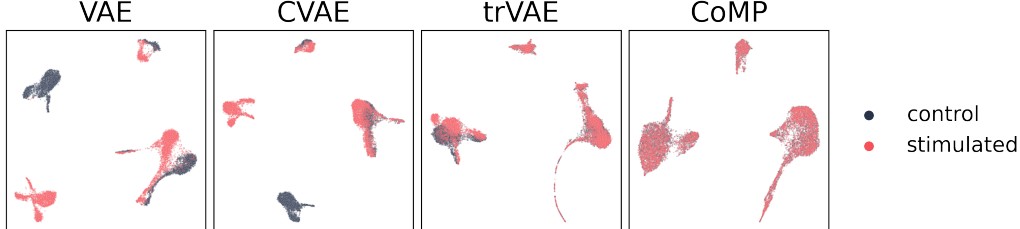

Figure 1: Latent representations of a single-cell gene expression dataset under two conditions (see Section 6.2). From fully disjointed (VAE) to a well-mixed pair of distributions (CoMP).

data space, methods that solve this problem can also be applied to counterfactual inference [10]. This same mathematical requirement for independence also occurs in the fair representation learning literature, in which we seek a representation that removes a sensitive attribute, e.g. gender.

Neither the VAE nor the conditional VAE (CVAE) [16] are typically successful at learning representations that achieve this desired independence, as shown in Figure 1. Despite the CVAE being theoretically able to remove batch effects, there is no constraint that prevents it from from separating different conditions in latent space. Existing methods use a penalty to encourage the CVAE to learn representations that overlap correctly in latent space, with Maximum Mean Discrepancy (MMD) [17] being the most common penalty, applied in the VFAE [18] and the more recent trVAE [10]. These methods, however, suffer from a number of drawbacks: conceptually, they introduce an extraneous discrepancy measure that is not a part of the variational inference framework; practically, they require the choice of, and hyperparameter tuning for, an MMD kernel; empirically, whilst trVAE is a significant improvement over an unconstrained CVAE, Figure 1 [trVAE] shows that it can still fail to exactly align different conditions in latent space.

To overcome these difficulties, we introduce *Contrastive Mixture of Posteriors (CoMP)*, a new method for learning aligned representations in a CVAE framework. Our method features the novel CoMP misalignment penalty that forces the CVAE to remove batch effects. Inspired by contrastive learning [19, 20], the penalty encourages representations from different conditions to be close, whilst representations from the same condition should be spread out. To achieve this, we approximate the requisite marginal distributions using mixtures of the variational posteriors themselves, leading to a penalty that does not require an extraneous discrepancy measure or a separately tuned kernel. We prove that the CoMP penalty is a stochastic upper bound on a weighted sum of KL divergences, so minimising the penalty minimises a well-established statistical divergence measure. We analyse the training gradients of the CoMP and MMD penalties, finding key differences that help explain why CoMP gradients are generally more stable and better suited to datasets with complex global structure.

As shown in Figure 1 [CoMP], our method can achieve visually perfect alignment on a number of real-world biological datasets. We apply CoMP to two challenging biological problems: 1) aligning gene expression profiles between tumours and their corresponding cell-lines, as tackled in [21] and 2) estimating the gene expression profile of an unperturbed cell as if it *had* been treated with a chemical perturbation (counterfactual inference) [9]. We show that CoMP outperforms existing methods, achieving state-of-the-art performance on both tasks. Finally, given the connections to fair representation learning, we apply CoMP to the problem of learning a representation that is independent of gender in the UCI Adult Income dataset [22], showing that we can learn a representation that is fully independent of the protected attribute whilst maintaining useful information for other prediction tasks. CoMP represents a conceptually simple and empirically powerful method for learning aligned representation, opening the door to answering high-value questions in biology and beyond.

## 2 Background

### 2.1 Variational Autoencoders and extensions

We begin by assuming that we have $n$ observations $\mathbf{x}_1, \ldots, \mathbf{x}_n$ of an underlying data distribution. Variational autoencoders (VAEs) [3, 4] explain the high-dimensional observations $\mathbf{x}_i$ using low dimensional representations $\mathbf{z}_i$. The standard VAE places a standard normal prior $\mathbf{z} \sim p(\mathbf{z})$ on

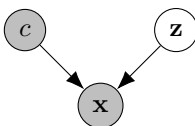

Figure 2: Structural Equation Model for observation $\mathbf{x}$ under known condition $c$ with unobserved latent variable $\mathbf{z}$. In this model, $\mathbf{z}$ and $c$ are independent in the prior.

the latent variable, and learns a generative model $p_\theta(\mathbf{x}|\mathbf{z})$ that reconstructs $\mathbf{x}$ using $\mathbf{z}$, alongside an inference network $q_\phi(\mathbf{z}|\mathbf{x})$ that encodes $\mathbf{x}$ to $\mathbf{z}$. Both $\theta$ and $\phi$ are trained jointly by maximising the ELBO, a lower bound on marginal likelihood given by $\log p_\theta(\mathbf{x}) \geq \mathbb{E}_{q_\phi(\mathbf{z}|\mathbf{x})}[\log p_\theta(\mathbf{x}|\mathbf{z})] -$ KL $[q_\phi(\mathbf{z}|\mathbf{x})\|p(\mathbf{z})]$. This can be maximised using stochastic optimisers [23, 3]. Various extensions of the VAE have been proposed, such as the $\beta$-VAE [24], which scales the KL term of the ELBO by a hyperparameter $\beta$. Because the isotropic normal prior may limit the expressivity of the model [25], various authors have considered alternative priors. For example, [26] proposed the Variational Mixture of Posteriors (VaMP) prior, that replaces the isotropic Gaussian with a mixture of posteriors from the encoder network itself, evaluated at a number of learned pseudo-inputs.

So far, we have assumed that the only data available are the observations $\mathbf{x}_1, \ldots, \mathbf{x}_n$, but in many practical applications we may have additional information such as a condition label for each observation. For example, in gene knock-out studies, we have information about which gene was targeted for deletion in each cell; in multi-batch experiments we have information about which experimental batch each samples was collected in. Thus, we augment our data by considering data pairs $(\mathbf{x}_1, c_1), \ldots, (\mathbf{x}_n, c_n)$ where $\mathbf{x}$ is the same high-dimensional observation, and $c$ is a label indicating the condition or experimental batch that $\mathbf{x}$ was collected under.

Whilst VAEs are theoretically able to model the pairs $(\mathbf{x}_i, c_i)$, it makes sense to build a model that explicitly distinguishes between the $\mathbf{x}$ and $c$. The simplest model is the Conditional VAE (CVAE) [16]. In this model, a conditional generative model $p_\theta(\mathbf{x}|\mathbf{z}, c)$ and a conditional inference network $q_\phi(\mathbf{z}|\mathbf{x}, c)$ are trained using a modified ELBO. A key observation for our work is that the CVAE has many different ways to model the data. For example, it can completely ignore the condition $c$ in $p_\theta$ and $q_\phi$, reducing to the original VAE. Assuming that $\mathbf{x}$ is not independent of $c$, this failure mode of the CVAE would be apparent on a visualization of the representations. For example, different values of $c$ might be visible as separate latent clusters, as shown in Figure 1 [CVAE].

## 2.2 Counterfactual inference

If $(\mathbf{x}_i, c_i)$ represents an RNA transcript and the gene knock-out applied to the cell, a natural question to ask is "How would the transcript have differed if a different knock-out $c'$ had been applied?" In general, *counterfactual inference* attempts to answer questions of the form "How would the data have changed if $c_i$ had been replaced by $c'$?" Answering counterfactual questions is a notoriously difficult task, because they naturally refer to unobservable data [27]. A principled approach to such questions is to adopt the framework of Structural Equation Models [28, 27]. For example, we could assume that the data generating process is given as in Figure 2. If this model is correct, counterfactual inference in the Pearl framework [27] can then be performed by: 1. *abduction*: inferring the latent $\mathbf{z}$ from $\mathbf{x}$ and $c$ using $p(\mathbf{z}|\mathbf{x}, c)$, 2. *action*: swap $c$ for $c'$, 3. *prediction*: use $p(\mathbf{x}|\mathbf{z}, c')$ to obtain a predictive distribution for the counterfactual. Thus, the counterfactual distribution of $\mathbf{x}_i$ observed with condition $c_i$ but predicted for condition $c'$ is given by

$$p\left(\mathbf{x}_{c=c'}|\mathbf{x}_i, c_i\right) = \int p(\mathbf{z}|\mathbf{x}_i, c_i)p(\mathbf{x}|\mathbf{z}, c')\, d\mathbf{z}. \tag{1}$$

In order to make use of this relationship, we must fit a latent variable model [29] such as a CVAE that will estimate the encoding distribution $p(\mathbf{z}|\mathbf{x}_i, c_i)$ and the generative distribution $p(\mathbf{x}|\mathbf{z}, c')$.

## 3 Unifying counterfactual inference, data integration and fairness

We have seen that batch effect correction, data integration and counterfactual inference are central problems of interest for the application of latent variable models in computational biology.

For counterfactual inference, latent variable models such as the CVAE are increasingly popular choices [29]. The failure mode in which different values of $c$ form separate latent clusters, however, can be catastrophic for this application. When this happens, simply switching $c_i$ to $c'$ is not correct, we have to account for the shift between clusters [9]. Mathematically, the latent space clustering phenomenon violates the assumption $\mathbf{z} \perp\!\!\!\perp c$ that is required by the model in Figure 2. Thus, whilst it is not always possible to know when we have found the right causal model [30], we can immediately say that a model in which $\mathbf{z}$ and $c$ are dependent is not correct.

Another key challenge for computational biology is data integration. Suppose our data $(\mathbf{x}_1, c_1), \ldots, (\mathbf{x}_n, c_n)$ in which $c_i$ indicates the experimental batch, exhibits *batch effects*—these are changes in the observation $\mathbf{x}_i$ due to the experimental conditions rather than true changes in the underlying biology. One approach to dataset integration is to create a representation $\mathbf{z} = \mathbf{z}(\mathbf{x}, c)$ that 'subtracts' the batch effects. Downstream tasks can then work with $\mathbf{z}$ in place of $\mathbf{x}$ without learning signal based on misleading batch effects. To know when we have successfully subtracted batch effects, we might assume that there are no population-level differences between batches. In other words, the marginal distribution of $\mathbf{z}$ should be the same for each value of the condition $c$.

Thirdly, this same notion of building a representation that cannot be used to recover $c$ has been studied widely in recent literature on fairness [31, 18, 32, 33]. In particular, if we wish to make a predictive rule based on $\mathbf{x}$ that does not discriminate between individuals in different conditions $c$, we can use a fair representation $\mathbf{z}$, one which cannot be used to recover $c$, as an intermediate feature and train our model using $\mathbf{z}$. Such a representation clearly needs to contain information from $\mathbf{x}$, but without containing any information that could be used to recover $c$.

To connect these three notions of 'alignment in representation space' we recall the key components of the CVAE—the encoder $q_\phi(\mathbf{z}|\mathbf{x}, c)$ and decoder $p_\theta(\mathbf{x}|\mathbf{z}, c)$—and we now drop the $\theta, \phi$ subscripts for conciseness. The marginal distribution of representations within condition $c \in \mathcal{C}$ is $q(\mathbf{z}|c) = \mathbb{E}_{p(\mathbf{x}|c)}[q(\mathbf{z}|\mathbf{x}, c)]$, and the marginal distribution of over all conditions not equal to $c$ is denoted

$$q(\mathbf{z}|\neg c) = \frac{\sum_{c' \in \mathcal{C}, c' \neq c} p(c') q(\mathbf{z}|c')}{\sum_{c' \in \mathcal{C}, c' \neq c} p(c')}. \tag{2}$$

The following Theorem brings together key notions in counterfactual inference, data integration and fair representation learning. See Appendix B for the proof.

**Theorem 1.** *The following are equivalent: 1)* $\mathbf{z} \perp\!\!\!\perp c$ *under distribution* $q$, *2) for every* $c, c' \in \mathcal{C}$, $q(\mathbf{z}|c) = q(\mathbf{z}|c')$, *3) for every* $c \in \mathcal{C}$, $q(\mathbf{z}|c) = q(\mathbf{z}|\neg c)$, *4) the mutual information* $I(\mathbf{z}, c) = 0$ *under distribution* $q$, *5)* $\mathbf{z}$ *cannot predict* $c$ *better than random guessing.*

# 4 Contrastive Mixture of Posteriors

We have seen that counterfactual inference, data integration and fair representation learning can be understood through the unified concept of learning a representation such that the latent variable $\mathbf{z}$ is independent of the condition $c$ under the distribution $q$, so that the latent clusters with different values of $c$ are perfectly aligned. Building off the CVAE, which rarely achieves this in practice, a number of authors have attempted to use a penalty term to reduce the dependence of $\mathbf{z}$ upon $c$ during training. The most successful methods, such as trVAE [10], are based on a Maximum Mean Discrepancy (MMD) [17]. We discuss this and other common methods in Section 5. Whilst trVAE and related methods can work well, they require an MMD kernel, not a part of the original model, to be specified and its parameters to be carefully tuned. Experimentally, we observe that MMD-based methods can often struggle when there is complex global structure in the latent space. We also analyse the gradients of MMD penalties, showing that they have several undesirable properties.

We propose a novel method to ensure the conditions of Theorem 1 do hold in a CVAE model. Our penalty is based on posterior distributions obtained from the model encoder itself. That is, we do not introduce any external discrepancy measure, rather we propose a penalty term that arises naturally from the model itself. Taking our inspiration from contrastive learning [19, 20] and the VaMP prior [26], we suggest a novel penalty to enforce equation condition 3) of Theorem 1. This equation requires the equality of the marginal distribution $q(\mathbf{z}|c)$ and $q(\mathbf{z}|\neg c)$ for each $c \in \mathcal{C}$. In practice, these marginal distributions can be approximated by finite *mixtures*. To encourage greater overlap between $q(\mathbf{z}|c)$ and $q(\mathbf{z}|\neg c)$, we can encourage points with the condition $c$ to be in areas of high density under the representation distribution for *other* conditions, i.e. areas in which $q(\mathbf{z}|\neg c)$ is also

high. To encourage this, we can add the penalty term $\mathcal{P}_0(\mathbf{z}_i, c_i) = -\log q(\mathbf{z}_i | \neg c_i)$ to the objective for the data pair $(\mathbf{x}_i, c_i)$. When we minimise $\mathcal{P}_0$, this brings the representations of samples under condition $c_i$ towards regions of high density under $q(\mathbf{z} | \neg c)$. Since the density $q(\mathbf{z} | \neg c)$ is not known in closed form, we approximate $q(\mathbf{z} | \neg c)$ using other points in the same training batch as $(\mathbf{x}_i, c_i)$. Indeed, suppose we have a batch $(\mathbf{x}_1, c_1), ..., (\mathbf{x}_B, c_B)$. We let $I_c$ denote the subset of indices for which $c_j = c$ and $I_{\neg c}$ denote its complement. We use the approximation

$$\log q(\mathbf{z}_i | \neg c_i) \approx \log \left( \frac{1}{|I_{\neg c_i}|} \sum_{j \in I_{\neg c_i}} q(\mathbf{z}_i | \mathbf{x}_j, c_j) \right) \tag{3}$$

and we will show in Theorem 2, this approximation in fact leads to a valid stochastic bound.

It may happen that the penalty $\mathcal{P}_0$ causes points to become too tightly clustered. Indeed, the penalty encourages latent variables to gravitate towards high density regions of $q(\mathbf{z} | \neg c_i)$. Inspired by contrastive learning, we include a second term which promotes higher entropy of the marginal, thereby avoiding tight clusters of points. Combined with $\mathcal{P}_0$, this leads us to a second penalty $\mathcal{P}_1(\mathbf{z}_i, c_i) = \log q(\mathbf{z}_i | c_i) - \log q(\mathbf{z}_i | \neg c_i)$. Again, the density $q(\mathbf{z} | c)$ is not known in closed form, but we can approximate it using points within the same training batch in a similar fashion to (3). Combining both approximations and taking the mean over the batch gives our *Contrastive Mixture of Posteriors (CoMP) misalignment penalty*

$$\text{CoMP penalty} = \frac{1}{B} \sum_{i=1}^{B} \log \left( \frac{1}{|I_{c_i}|} \sum_{j \in I_{c_i}} q(\mathbf{z}_i | \mathbf{x}_j, c_i) \right) - \log \left( \frac{1}{|I_{\neg c_i}|} \sum_{j \in I_{\neg c_i}} q(\mathbf{z}_i | \mathbf{x}_j, c_j) \right). \tag{4}$$

where $\mathbf{x}_{1:B}, c_{1:B}, \mathbf{z}_{1:B} \sim \prod_{i=1}^{B} p(\mathbf{x}_i, c_i) q(\mathbf{z}_i | \mathbf{x}_i, c_i)$ is a random training batch of size $B$, $I_c$ denotes the subset of $\{1, \ldots, B\}$ with condition $c$ and $I_{\neg c} = \{1, \ldots, B\} \setminus I_c$. Our method therefore utilises a training penalty for CVAE-type models that encourages the conditions of Theorem 1 to hold by using mixtures of the variational posteriors themselves to approximate $q(\mathbf{z} | c)$ and $q(\mathbf{z} | \neg c)$. We do not introduce an additional kernel or hyperparameter-heavy discrepancy measures.

As hinted at by the definition of $\mathcal{P}_1$, CoMP can be seen as approximating a symmetrised KL-divergence between the distributions $q(\mathbf{z} | c)$ and $q(\mathbf{z} | \neg c)$. In fact, the following theorem shows that the CoMP misalignment penalty is a *stochastic upper bound on a weighted sum of KL-divergences*.

**Theorem 2.** *The CoMP misalignment penalty satisfies*

$$\mathbb{E}_{\prod_{i=1}^{B} p(\mathbf{x}_i, c_i) q(\mathbf{z}_i | \mathbf{x}_i, c_i)} \left[ \frac{1}{B} \sum_{i=1}^{B} \log \left( \frac{1}{|I_{c_i}|} \sum_{j \in I_{c_i}} q(\mathbf{z}_i | \mathbf{x}_j, c_i) \right) - \log \left( \frac{1}{|I_{\neg c_i}|} \sum_{j \in I_{\neg c_i}} q(\mathbf{z}_i | \mathbf{x}_j, c_j) \right) \right]$$
$$\geq \sum_{c \in \mathcal{C}} p(c) \, \text{KL} \left[ q(\mathbf{z} | c) || q(\mathbf{z} | \neg c) \right]$$

*and the bound becomes tight as $B \to \infty$.*

The proof is presented in Appendix B. Our result reveals that our new penalty directly enforces condition 3) of Theorem 1 by reducing the KL divergence between each pair $q(\mathbf{z} | c), q(\mathbf{z} | \neg c)$ weighted by $p(c)$. As with standard contrastive learning, our method benefits from larger batch sizes. We add the CoMP misalignment penalty to the familiar $\beta$-VAE objective to give our *complete training objective* for a batch of size $B$ as

$$\mathcal{L}_B^{\text{CoMP}} = \frac{1}{B} \sum_{i=1}^{B} \left[ \log p(\mathbf{x}_i | \mathbf{z}_i, c_i) + \beta \log \frac{p(\mathbf{z}_i)}{q(\mathbf{z}_i | \mathbf{x}_i, c_i)} - \gamma \log \left( \frac{\frac{1}{|I_c|} \sum_{j \in I_c} q(\mathbf{z}_i | \mathbf{x}_j, c_i)}{\frac{1}{|I_{\neg c}|} \sum_{j \in I_{\neg c}} q(\mathbf{z}_i | \mathbf{x}_j, c_j)} \right) \right] \tag{5}$$

with one new hyperparameter $\gamma$ that controls the strength of the regularisation we apply to enforce the requirements $\mathbf{z} \perp\!\!\!\perp c$. Theorem 2 shows that, if $\mathcal{L}_B^{\beta}$ is the standard $\beta$-VAE objective, then we are maximising $\mathbb{E}\left[ \mathcal{L}_B^{\text{CoMP}} \right] \leq \mathbb{E}\left[ \mathcal{L}_B^{\beta} \right] - \gamma \sum_{c \in \mathcal{C}} p(c) \, \text{KL} \left[ q(\mathbf{z} | c) || q(\mathbf{z} | \neg c) \right]$.

## 4.1 Analysing CoMP gradients

Before presenting empirical results on the performance of CoMP, we attempt to understand how it differs from existing penalties in the literature. Specifically, we compare CoMP with a Gaussian

posterior family with MMD using a Radial Basis Kernel [34]. In Appendix C, we show that both methods can be interpreted as applying a penalty to each element $\mathbf{z}_i, c_i$ of the training batch. We show further that, under certain conditions, the gradient of the MMD penalty for $\mathbf{z}_i, c_i$ takes the form

$$\nabla_{\mathbf{z}_i} \mathcal{P}_{\text{MMD}}(\mathbf{z}_i, c_i) = \frac{2}{|I_{c_i}|^2} \sum_{j \in I_{c_i}} e^{-\|\mathbf{z}_i - \mathbf{z}_j\|^2} (\mathbf{z}_j - \mathbf{z}_i) - \frac{4}{|I_{\neg c_i}||I_{c_i}|} \sum_{j \in I_{\neg c_i}} e^{-\|\mathbf{z}_i - \mathbf{z}_j\|^2} (\mathbf{z}_j - \mathbf{z}_i), \quad (6)$$

whilst the CoMP penalty gradient takes the form

$$\nabla_{\mathbf{z}_i} \mathcal{P}_{\text{CoMP}}(\mathbf{z}_i, c_i) = \frac{2 \sum_{j \in I_{c_i}} e^{-\|\mathbf{z}_i - \mu_{\mathbf{z}_j}\|^2} (\mu_{\mathbf{z}_j} - \mathbf{z}_i)}{B \sum_{j \in I_{c_i}} e^{-\|\mathbf{z}_i - \mu_{\mathbf{z}_j}\|^2}} - \frac{2 \sum_{j \in I_{\neg c_i}} e^{-\|\mathbf{z}_i - \mu_{\mathbf{z}_j}\|^2} (\mu_{\mathbf{z}_j} - \mathbf{z}_i)}{B \sum_{j \in I_{\neg c_i}} e^{-\|\mathbf{z}_i - \mu_{\mathbf{z}_j}\|^2}} \quad (7)$$

where $\mu_{\mathbf{z}_j}$ is the variational mean for $\mathbf{z}_j$. One important feature of the MMD gradients is that, if $\|\mathbf{z}_i - \mathbf{z}_j\|^2$ is large for all $j \neq i$, for instance when the point $\mathbf{z}_i$ is part of an isolated cluster, then the gradient to update the representation $\mathbf{z}_i$ will be small. So if $\mathbf{z}_i$ is already very isolated from the distribution $q(\mathbf{z}|\neg c_i)$, then the gradients bringing it closer to points with condition $\neg c_i$ will be small. In comparison to the MMD gradient, it can be seen that gradients for CoMP are *self-normalised*. This means that the gradient through $\mathbf{z}_i$ will be large, even when $\mathbf{z}_i$ is very far away from any points with condition $\neg c_i$. This, in turn, suggests that that CoMP is likely to be preferable to MMD when we have a number of isolated clusters or interesting global structure in latent space, something which often occurs with biological data. The CoMP approach also bears a relationship with nearest-neighbour approaches [35]. Indeed, for a Gaussian posterior as $\sigma \to 0$, the $\neg c_i$ term of the gradient places all its weight on the nearest element of the batch under condition $\neg c_i$.

## 5   Related Work

The problem of batch correction in data integration has been addressed using linear [12, 13] and nonlinear methods [14, 15] that perform transformations of the original feature space. In both cases, the goal is to transform the feature space so that information related to the scientific question of interest is retained while dependence on the batch (or nuisance covariate) is reduced. Methods based on representation learning learn a low-dimensional representation, $\mathbf{z} = q(\mathbf{x})$, which is independent of nuisance factors while also being a faithful representation of the original data [18, 5, 36, 10, 37]. Of these, the work that is most similar to ours are the VFAE [18], in which the authors introduce an MMD [17] penalty to encourage the marginal distributions of $\mathbf{z}$ under different values of $c$ to be close, and the trVAE [10], where the MMD penalty is applied to the output of the first layer of the decoder, rather than to $\mathbf{z}$ directly. Representation learning algorithms for counterfactual inference have been shown to benefit from a penalty enforcing distributional similarity between the representations of the treated and untreated samples [12]. Elsewhere, authors have applied the variational autoencoder to inference on causal graphs [38, 39, 40].

## 6   Experiments

We perform experiments on three datasets; 1) Tumour / Cell Line: bulk expression profiles of tumours and cancer cell-lines across 39 different cancer types; 2) Single-cell PBMCs: single-cell gene expression (scRNA-seq) profiles of interferon (IFN)-$\beta$ stimulated and untreated peripheral blood mononuclear cells (PBMCs) [41]; 3) UCI Adult Income: personal information relating to education, marriage status, ethnicity, self-reported gender of census participants and a binary high / low income label ($50,000 threshold) [22]. All experiments used a 90/10 training/validation split.

The two broad objectives across our experiments are 1) to demonstrate the extent to which the two random variables $\mathbf{z}_i$ and $c_i$ are independent, and 2) to quantify useful information retained in $\mathbf{z}_i$. To benchmark CoMP on the first objective, we use the following pair of $k$ nearest-neighbor metrics: kBET$_{k,\alpha}$ [42], the metric used to evaluate batch correction methods in biology, and a local Silhouette Coefficient [43] $s_{k,c}$. In both cases a low value close to zero would indicate good local mixing of sample representations. As for the second objective, if we assume the existence of an additional discrete label $d_i$ that represents information one wishes to preserve – in the Tumour / Cell Line case, $d_i$ is the cancer type, while for the PBMC experiment, it refers to cell type – then we calculate kBET and $s$ separately for every fixed-$d_i$ subpopulation and take the mean. We refer to these as the *mean Silhouette Coefficient* $\tilde{s}_{k,c}$ and the *mean kBET* metric m-kBET respectively. Full details of the datasets and metrics are given in Appendix D.

Table 1: Tumour / Cell Line experiment results, with $k = 100$, $c$ = Cell Line, and $\alpha = 0.01$. $s_{k,c}$ and $\tilde{s}_{k,c}$ are the two Silhouette Coefficient variants (see Section 6). The top scores are in **bold**.

| | Accuracy | $s_{k,c}$ | $\text{kBET}_{k,\alpha}$ | $\tilde{s}_{k,c}$ | $\text{m-kBET}_{k,\alpha}$ |
|---|---|---|---|---|---|
| VAE | 0.209 | 0.658 | 0.974 | 0.803 | 0.581 |
| CVAE | 0.328 | 0.554 | 0.931 | 0.684 | 0.571 |
| VFAE | **0.585** | 0.168 | 0.258 | 0.198 | 0.188 |
| trVAE | **0.585** | 0.096 | 0.163 | 0.138 | 0.123 |
| Celligner | 0.578 | 0.082 | 0.525 | 0.568 | 0.226 |
| *CoMP (ours)* | 0.579 | **0.023** | **0.160** | **0.094** | **0.101** |

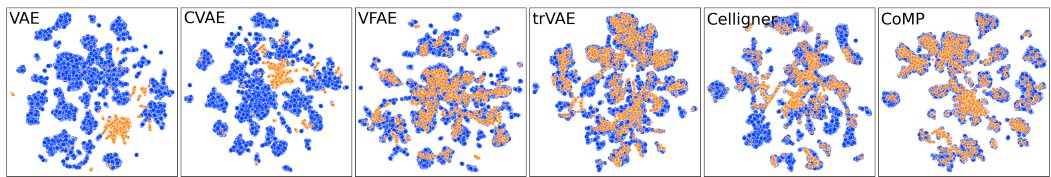

Figure 3: 2D UMAP projection of posterior means of $\mathbf{z}_i$ from Tumour / Cell Line data. Tumours (blue) and cell lines (orange).

## 6.1 Alignment of tumour and cell-line samples

Despite their widespread use in pre-clinical cancer studies, cancer cell-lines are known to have significantly different gene expression profiles compared to their corresponding tumour samples. Here we evaluate the ability of CoMP to factorise out the tumour / cell line condition from its latent representations. This can be seen as both a dataset integration and batch effect correction task. In addition to the set of $k$ nearest neighbor-based mixing evaluations, we train a Random Forest model on the representations of the tumour samples and their cancer-type labels and assess the prediction accuracy on held-out cell lines. To match the results from [21], the evaluations are performed on the 2D UMAP projections, The results are presented in Table 1.

As expected, both the VAE and CVAE baselines fail at the mixing task; the three explicitly penalised VAE models and, to a lesser extent, the *Cellinger* method have good mixing performances, with CoMP outperforming the benchmark models by a significant margin on the silhouette coefficient and kBET metric, while successfully maintaining a high accuracy in the cancer-type prediction task. We also see from Figure 3 that CoMP representations have the fewest instances of isolated tumour-only clusters. Finally, from our evaluation on the $\tilde{s}$ and m-kBET metrics, we can deduce that the occurrence of cell lines of one cancer type erroneously clustering around tumours of a different type is less frequent for CoMP compared to the other models. In Appendix D we qualitatively validate this for several example clusters.

## 6.2 Interventions

Obtaining molecular measurements from biological tissues typically requires destructive sampling. For example, to obtain scRNA-seq data, each cell is lysed so that the RNA molecules contained within it can be extracted and sequenced. This process destroys each cell, meaning that we are unable to study the gene expression profile of the same cell over time or under multiple experimental conditions. As we discussed in Section 2.2, counterfactual inference can be used to predict how the molecular status of a destroyed biological sample would have differed if it were measured under different experimental conditions, such as applications of different drugs.

To assess CoMP's utility in counterfactual inference, we trained it on scRNA-seq data from PBMCs that were either stimulated with IFN-$\beta$ or left untreated (control) [41]. It is clear from Figure 4 that IFN-$\beta$ stimulation causes clear shifts in the latent space between stimulated and control cells from the same cell type. Noticeably, the CD14 and CD16 monocyte and dendritic cell (DC) populations see greater shifts in their gene expression after stimulation. CVAE fails to align these particular cell types in the latent space, while trVAE, VFAE and CoMP perform better. However, stimulated and control

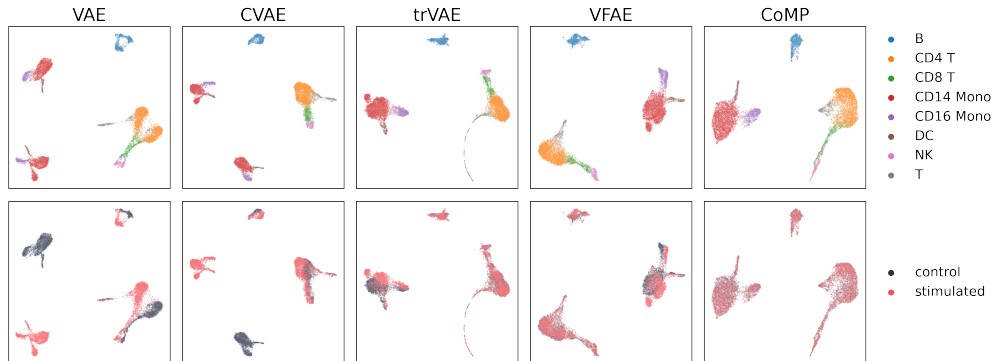

Figure 4: 2D UMAP projections of posterior means of $\mathbf{z}_i$ derived from stimulated and control PBMC scRNA-seq data. Top row: colours indicate immune cell types, bottom row: colours indicate condition (IFN-$\beta$ stimulation or control).

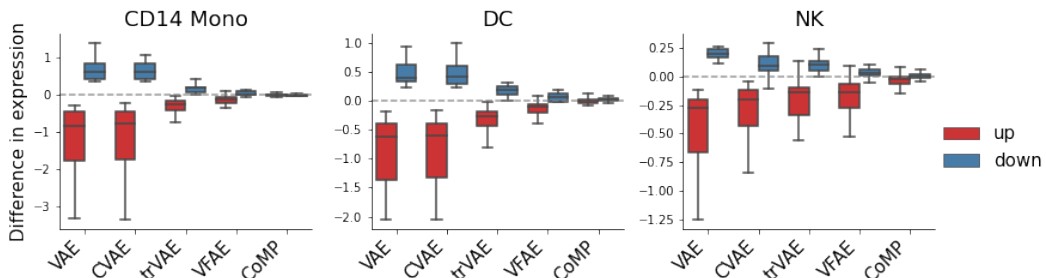

Figure 5: The difference in gene expression values for the top 50 differentially expressed genes (up-regulated: red, down-regulated: blue) between IFN-$\beta$ stimulated cells and counterfactually stimulated control cells for CD14 monocytes, dendritic cells (DC) and natural killer (NK) cells. See Appendix D for further details.

cells are better mixed in the latent space derived from CoMP than those from the other models (see metrics presented in Appendix D).

Next we perform a counterfactual prediction task under a IFN-$\beta$ control-to-stimulation variable swap, i.e. the gene expression profiles for control cells were reconstructed through the decoder with the condition, $c \to$ stimulated. This means we utilise equation (1) with our encoder $q_\phi(\mathbf{z}|\mathbf{x}, c)$ and decoder $p_\theta(\mathbf{x}|\mathbf{z}, c')$ in place of $p(\mathbf{z}|\mathbf{x}, c)$ and $p(\mathbf{x}|\mathbf{z}, c')$. The degree to which the models respect the requirement $\mathbf{z} \perp\!\!\!\perp c$ will influence the quality of predictions. Figure 5 shows how the profiles of (actual) stimulated cells differ from the counterfactual predictions for a selection of cell types (see Appendix D for the complete set of results). We see that baseline models tend to systematically underestimate the expression of genes up-regulated by stimulation and overestimate those down-regulated. CoMP outperforms all other models by accurately predicting the expression alterations brought about by stimulation.

## 6.3 Fair Classification

The goal for this fair classification task is to learn a representation on the Adult Income dataset that is not predictive of an individual's gender whilst still being predictive of their income. We compute a baseline by predicting gender and income labels directly from the input data and compare our method to the published results for the VFAE [18] and the trVAE. We also include results for a standard VAE and CVAE. Unlike in [18], where the representations $\mathbf{z}$ are sampled from the posterior before classification, our experiments used the posterior means to avoid the noise from sampling acting to mask the inclusion of predictive information about gender in the encodings.

CoMP achieves a gender accuracy that is close to random (67.5%), tying with the VFAE results from [18] whilst also remaining competitive with the other methods on income accuracy (Table 2). CoMP

Table 2: UCI Adult Income experiment results with $k = 1000$, $c = $ Male for $s_{k,c}$, and $k = 100$, $\alpha = 0.01$ for kBET$_{k,\alpha}$. A lower gender prediction accuracy is better; 0.675 is the lowest achievable.

|  | Gender Acc. | Income Acc. | $s_{k,c}$ | kBET$_{k,\alpha}$ |
| --- | --- | --- | --- | --- |
| Original data | 0.796 | **0.849** | 0.067 | 0.786 |
| VAE | 0.764 | 0.812 | 0.054 | 0.748 |
| CVAE | 0.778 | 0.819 | 0.054 | 0.724 |
| VFAE (sampled) [18] | 0.680 | 0.815 | - | - |
| VFAE (mean) | 0.789 | 0.805 | 0.046 | 0.571 |
| trVAE | 0.698 | 0.808 | 0.066 | 0.731 |
| *CoMP (ours)* | **0.679** | 0.805 | **0.011** | **0.451** |

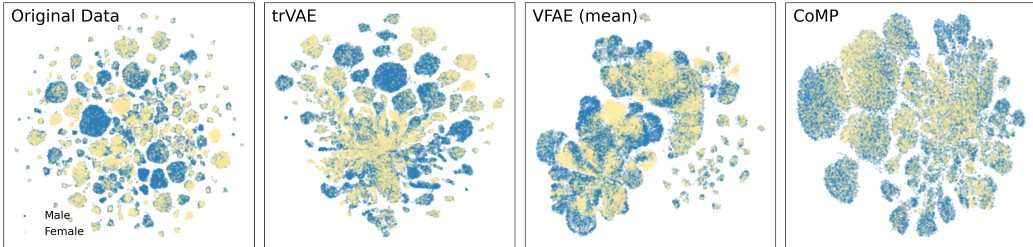

Figure 6: UMAP projections for the UCI Adult Income dataset, coloured by gender. Showing the original data and latents for trVAE, VFAE and CoMP. Male (blue) and female (yellow).

also outperforms all methods on the nearest neighbour and silhouette metrics (Table 2). Latent space mixing between males and females can be seen qualitatively in the 2D UMAP projection (Figure 6).

## 7   Conclusion

**Limitations**   We presented Contrastive Mixture of Posteriors (CoMP) as an effective means to perform batch correction, data integration, counterfactual inference and fair representation learning in a CVAE framework. Whilst CoMP covers the majority of common use-cases for these tasks, there are several limitations that are avenues of future research. For example, in scRNA-seq analysis, there is often the need to integrate more than two datasets together, or to adjust for continuous condition variables. Mathematically, CoMP is applicable to any number of discrete conditions, and it would be interesting to apply it to a setting with $> 2$ conditions. Extensions of CoMP could tackle the case of a continuous condition variable. Additionally, CoMP requires the condition variable $c$ to be fully observed: future work might attempt to generalise to the partially observed case.

**Summary**   We identified marginal independence between the representation $\mathbf{z}$ and condition $c$ as the mathematical thread linking data integration, counterfactual inference and fairness. We proposed CoMP, a novel method to enforce this independence requirement in practice. We saw that CoMP has several attractive theoretic properties. First, CoMP only uses the variational posteriors, requiring no additional discrepancy measures such as MMD. Second, we proved that the CoMP penalty can be interpreted as an upper-bound on a weighted sum of KL divergences, connecting it to a well-founded divergence measure. Third, we demonstrated that, unlike MMD, CoMP gradients have a self-normalising property, allowing one to obtain strong gradients for distant points in a latent space with complex global structure. Empirically, we demonstrated CoMP's performance when applied to two biological and one fair representation learning dataset. These biological datasets are of critical importance in drug discovery, for example matching cell-lines to tumours for effective pre-clinical assay development of anti-cancer compounds. Overall, CoMP has the best in class performance on all tasks across a range of metrics that measure either latent space mixing or fairness.

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
