# OpenReview forum: "Contrastive Mixture of Posteriors for Counterfactual Inference, Data Integration and Fairness"
_NeurIPS.cc/2021/Conference — NeurIPS 2021 Submitted_

### Official Review · Reviewer_UbUq · 2021-07-06

**Rating:** 7
**Confidence:** 3

**Summary:**

This paper considers the problem of data integration, counter factual inference and fairness concentrating on large data sets where each data point lives in high dimensions. The authors generate an embedding of the data into a latent space in such a way that the distribution of the embeddings is the same for data from all clusters. The authors propose to learn such an embedding through a variation autoencoder (VAE). They come up with a novel penalty called contrastive mixture of posteriors misalignment (CoMP) penalty which they use to with the ELBO loss to train the VAE. They show empirical results chiefly on single-cell RNA-seq data and also on one income data-sets.

**Limitations And Societal Impact:**

listed above

**Main Review:**

The authors consider the problems of data integration, counter factual inference and fairness. They primarily concentrate on single cell RNA-seq for the first two points - a new technology in which one can obtain gene expression on a cell by cell basis  giving us data-sets with millions of cells - and tens of thousand measurements of gene expression for each cell. However different experiments carried out by different scientists unfortunately have batch effects. Being able to combine data from experiments run different scientists to do analysis is necessary for scientific integration - making data integration and dealing with batch effects important problems to consider. The counter-factual inference in single cell datasets is less important as there usually are technical and biological replicates available for experiments. There of course is a plethora of literature on fairness. I did think that the connection between the three was nice, even though it is obvious.

The authors consider the method of embedding data from different batches into the same latent space as a means of getting rid of batch effects. However, this approach is explicitly not suitable for data integration in single cell RNA-seq because the data generated by different scientists is usually not from the exact same tissue from the same individual, and thus one would expect  the distribution of the latent space to be different in different batches even if there was no batch effect (without additioinal information the problem is not even well defined - but that's what makes working with single cell RNA-seq hard and interesting).

The authors consider a VAE as means of obtaining such an embedding. They do this by coming up with a novel penalty to the ELBO loss. The penalty penalises differences in the distribution of points in the latent space for different batches - and is different from existing VAE based penalties in that it uses the posterior of the model learnt in the penalty rather than a different kernel.

The authors then provide empirical results on Single Cell RNA seq and an income data. One concern I have is that the authors limit their comparisons to VAE based methods for Single Cell RNA-seq when they are not the most popular ways in literature to deal with batch effects and counter-factual inference not comparing with methods like those of [14], and many other papers exploring that problem such as [100], [101], and [102]. Similarly, there is no comparison for non-VAE based methods (for example [103]) for fairness claims.

The paper is well written and I liked the broad direction of the paper. But the formulation does not match the application of single cell RNA-seq that the paper purports to target. Furthermore, empirical results are found wanting comparing it with non VAE based approaches; which makes its contribution from the application perspective quite small. The only novelty in the paper seems to be using a different penalty in training the VAE - and I dont know the VAE literature enough to know who important or novel a contribution that is.

To me this is a borderline paper leaning towards rejection from an applications perspective. I like the idea and it is well written, but the formulation does not target any problem which analysts would be interested in and empirical results are not thorough.


[100] Korsunsky I, Millard N, Fan J, Slowikowski K, Zhang F, Wei K, Baglaenko Y, Brenner M, Loh P-r, Raychaudhuri S. Fast, sensitive and accurate integration of single-cell data with Harmony. Nature Methods

[101] Welch J, Kozareva V, Ferreira A, Vanderburg C, Martin C, Macosko E. Integrative inference of brain cell similarities and differences from single-cell genomics. bioRxiv. 2018:459891 Available from: http://biorxiv.org/content/early/2018/11/02/459891.abstract

[102]Stuart T, Butler A, Hoffman P, Hafemeister C, Papalexi E, Mauck WM 3rd, et al. Comprehensive integration of single-cell data. Cell. 2019;177:1888–1902.e21.

[103]Agarwal, Alekh, Alina Beygelzimer, Miroslav Dudík, John Langford, and Hanna Wallach. "A reductions approach to fair classification." In International Conference on Machine Learning, pp. 60-69. PMLR, 2018.


**Time Spent Reviewing:**

4.5

---

> ### Author Response · Authors · 2021-08-10
> **Individual reply to Reviewer UbUq**
>
> Thank you for your thoughtful review. We are happy to see that you appreciated our exposition of the connection between batch correction, counterfactual inference and fairness, and that you felt the paper was well written.
>
> > this approach is explicitly not suitable for data integration in single cell RNA-seq
>
> > the formulation does not match the application of single cell RNA-seq
>
> We disagree. As pointed out by reviewer SVjF, the assumption that $z$ and $c$ are independent is a strong one, and could be violated if there are phenotypical differences between batches. However, this is very far from invalidating our approach.
>
> First, we agree further discussion on this point would improve the paper, and we will make sure to add this.
>
> Second, as well as the data integration experiment and the counterfactual inference experiment on single-cell RNA-seq data presented in our submitted paper, to directly address your concerns we decided that it would be best to validate empirically that CoMP can be used in the specific case of batch correction for single cell RNA-seq data. See the comment to all reviewers for details and results from this new experiment.
>
> Third, as you stated, a mathematical assumption is required unless there are explicit biological markers that can differentiate between phenotypical and artificial batch effects. We believe that we have simply been more explicit and more open about the assumption than previous papers, not that we are unusual in making assumptions of this kind.
> For example, many batch correction and/or data integration methods that are widely used in both the scRNA-seq and bulk RNA-seq community, make strong assumptions. For example, models such as Probabilistic Estimation of Expression Residuals (PEER) [1] or Surrogate Variable Analysis (SVA) [2] are latent factor models. These models estimate batch effects, or any technical confounding, as latent factors, in which the batches/technical biases aren't observed but are estimated from the data. These models all assume that latent batch effects are independent from any biological variation as they are subsequently regressed out of the data. In addition, TrVAE, conditional scVI and scMVAE, a model specifically designed for data integration across modalities (scRNA-seq and single-cell chromatin accessibility), all use a conditional VAE framework, so similar assumptions are also implicit in their work. So whilst these assumptions are not strictly true in practice, these methods have all been shown to have excellent empirical performance, including CoMP.
>
> Fourth, there are both theoretical and practical reasons to use CoMP, even when we do not believe that $z$ and $c$ should be exactly independent. Practically, we can change the coefficient gamma in equation (5). When this is small, the model will focus more on providing accurate representations, and representations will not end up perfectly aligned, instead the distance (formally, weighted KL divergence) between different conditions will be reduced. This can be made rigorous by viewing gamma as a Lagrange multiplier. From this perspective, minimising our loss in (5) amounts to solving a constrained optimisation problem with a constraint $\sum_c p(c) KL(q(z|c)||q(z|\neg c)) \le K_0$, for some constant $K_0$. Thus, CoMP can be interpreted as ensuring that there is some level of alignment between different values of c (via the constraint on the KLs) without requiring total independence. We will be sure to add this discussion and justification to the paper, and to talk more broadly about how the independence assumption could break down in the batch correction setting.
>
> >  the authors limit their comparisons to VAE based methods for Single Cell RNA-seq
>
> This is incorrect. In our ‘Alignment of tumour and cell-line samples’ (see Table 1 and Sec 6.1), we compared against the non-VAE method from Celligner. Celligner uses mutual nearest neighbours (MNN), explicitly wrapping the function from the Seurat library, one of the standard methods used in the single cell literature.
>
> To further address your concerns, in our new experiment doing batch correction for single cell RNA-seq data, we compared CoMP against both the Seurat and Harmony baselines. Our metrics show that CoMP not only gets better overall alignment, but also that, restricting to each cell subtype in turn, CoMP correctly aligns matching cells of the same subtype from different batches. See the response to all reviewers for full details.
>
> > The only novelty in the paper seems to be using a different penalty in training the VAE
>
> The paper offers a number of novelties beyond the specific training method for the CVAE. We formally discuss counterfactual inference, batch correction and fairness. We believe that the reviewer discussion around the mathematical assumptions that go into batch correction actually highlights that this presentation is new and stimulating, and can help researchers to be explicit about, and to refine the assumptions behind data integration methods.
>
> It is not just the CoMP penalty that constitutes a novelty for training the CVAE, but also our theoretical analysis that links this to the weighted sum of KLs (Theorem 1) thereby providing a solid justification for this method. We also provided a theoretical discussion.
>
> On the application side, we believe that the experiments we have presented, along with our new experiment doing batch correction on single cell RNA-seq data, demonstrate very clearly that our method can perform well on several key tasks in computational biology, and that there are applications beyond biology (fair representations).
>
> > The counter-factual inference in single cell datasets is less important as there usually are technical and biological replicates available
>
> One can always generate counterfactual data if you have additional biological replicates, but sequencing is expensive and there are many real-world situations where samples are limited, so being able to generate counterfactual data in-silico can be useful. In some other situations, it may not be possible to generate real data to answer a counterfactual question. For example, in the field of precision medicine, we may be interested in knowing what the gene expression of a set of specific cells, observed in patient A, might have been if they had instead come from patient B.
>
> **References**
>
> [1] Stegle, Oliver, Leopold Parts, Matias Piipari, John Winn, and Richard Durbin. "Using probabilistic estimation of expression residuals (PEER) to obtain increased power and interpretability of gene expression analyses." Nature protocols 7, no. 3 (2012): 500-507.
>
> [2] Leek, Jeffrey T., and John D. Storey. "Capturing heterogeneity in gene expression studies by surrogate variable analysis." PLoS genetics 3, no. 9 (2007): e161.

---

> > ### Comment · Reviewer_UbUq · 2021-08-22
> > **Thank you**
> >
> > Thanks you for your response. I'm quite happy to update my score to a 7.

---

### Official Review · Reviewer_SVjF · 2021-07-15

**Rating:** 8
**Confidence:** 4

**Summary:**

The authors propose a penalty term called Contrastive Mixture of Posteriors (CoMP) for conditional variational autoencoders to learn latent representations invariant of a condition. Their penalty term doesn’t have extraneous requirements and is free from requiring a kernel (as is the case case with MMD) and enforces independence in latent space with respect to a condition (e.g. batch effect, sensitive information etc.). CoMP penalty itself is made up of two components which 1) encourage latent representations to be clustered together in both conditions - under a condition and under complement of said condition, 2) avoid tight clustering using a contrastive learning approach.

The work is useful in some computational biology tasks (batch correction, counterfactual inference) which require low-dimensional representations of high-dimensional sequencing data free from batch effect or other conditions, and also in fair classification to learn a classifier free from sensitive demographic information. For each of these tasks, the authors have presented one experiment and have compared their methods with existing methods based on variational and conditional variance autoencoders.

**Limitations And Societal Impact:**

While I think the most important limitation of this work (the need for independence between z and c) was not mentioned, some other limitations were briefly (but appropriately) discussed in the conclusion.
The societal impact were not really discussed but I am not sure how relevant it would be for this specific paper.

**Main Review:**

The paper is very well-written and easy to follow. The applications and tasks considered are well-motivated with Theorem 1. While relatively straightforward, I find the connections drawn between data integration, counterfactual inference and fair classification to be very useful as it brings together references and results from sub-fields that might not be very aware of each other. Finally, the proposed method is very relevant and well-established for both domains considered in the paper - computational biology and fair classification. The limitations mentioned are also well thought out.

My main remarks (roughly in decreasing order of importance) are:

1) Assuming independence between z and c is absolutely crucial to the foundation of the studied setting, the correctness of the suggested method and to its theoretical analysis. However, the assumption rarely holds with biological data where different samples might, at the same time, present different phenotypes, technical biases (or batch effects), cell type compositions etc. I would be very curious to see results on the integration of datasets where we know the assumption is broken. For instance, if one cell type is present only in one batch, will their latent codes overlap with cells from a different cell type from the other batch? This issue, usually refered to as "over-correction" is absolutely crucial in practice. At least in the discussion, I think it could be nice to comment on it and give some insights on how this issue could be tackled both in theory and in practice.

2) The batch correction experiment considered is for alignment of tumor RNAseq (TCGA) and cell-lines (Cellinger). For a proper evaluation of batch correction, further experiments especially on single-cell data, where batch effects are much more prominent and the task is generally harder, should have been considered.

3) The only batch correction method evaluated outside of variational inference based approaches is Cellinger. It would be good to compare the model on single cell data with at least some popular (among single-cell practitioners) batch correction methods in use, e.g. Seurat and Harmony.

4) While understandably, as argued in the paper, the goal is to learn meaningful latent representations free from batch effect, it would still be interesting to evaluate batch correction in the original gene space (Especially considering that MSEs of reconstructions for counterfactual inference experiment are already shown in Tables 9-12 in Appendix D). This would also provide more opportunities for downstream analysis and evaluation of corrected data.

5) Only prostate cancer clusters have been visually shown in detail in Fig. 7 of Appendix D for trVAE and CoMP. It would be good to have detailed visualizations for more cancer clusters, or at least to include other methods for prostate cancer clusters. This is particularly important as in Section 6.1, it is mentioned that Appendix D includes qualitative results for several clusters.

6) In both batch correction and counterfactual inference experiments, the UMAPs of original data are not shown (Only the latent representations in Figures 3 and 4.). While it is clear that original data would be clustered by condition (batch or original and stimulated), it would still be useful to see that.

7) For fair classification, the authors mentioned that the paper shows “published results” from trVAE and VFAE, however in the original trVAE, this experiment was not performed. For completeness, it will be good to include the source where those results were published.

-------------
After checking the author's response, I am upgrading my rating from 7 to 8.

**Time Spent Reviewing:**

5

---

> ### Author Response · Authors · 2021-08-10
> **Individual reply to Reviewer SVjF**
>
> Thank you for a thoughtful and detailed review. We are extremely pleased to hear that you found the paper well written and easy to follow, and considered our method very relevant and well-established for the domains that we consider in the paper. To go over your points in order:
>
> **1.** This is a really important point, and one that we will add further discussion on in the paper. It’s absolutely correct that the requirement of exact independence between $z$ and $c$ might not match a batch correction problem in which there are both experimental batch and phenotypical differences present between batches. The first thing to say is that some mathematical assumption is needed here in the absence of a rule for distinguishing between phenotypical and artificial differences. We believe that our assumption has actually been used implicitly by many previous authors (e.g. in latent factor model-based approaches like surrogate variable analysis [1]). We are perhaps the first to make it explicit.
>
> Second, there are practical and theoretical ways to try and reduce “over-correction”. One approach is to focus on the weight gamma in equation (5) that scales the CoMP penalty. When this is small, the model will focus more on providing accurate representations, and the “force” to perfectly align representations will be smaller. This can be justified more rigorously by viewing gamma as a Lagrange multiplier. From this perspective, minimising our loss in (5) amounts to solving a constrained optimisation problem with a constraint $\sum_c p(c) KL(q(z|c)||q(z|\neg c)) \le K_0$, for some constant $K_0$. That means that the CoMP method can be interpreted as ensuring that there is **some** level of alignment between different values of $c$ (via the constraint on the KLs), but it doesn’t require total independence.
>
> Lastly, our mean-kBET and mean-silhouette scores are designed to check that CoMP is correctly aligning matching cell types from different batches- empirically we find that it does do this. In general, these scores could be helpful to diagnose over-correction.
>
> **2/3.** These are two good points, and we felt directly addressing them with a new experiment would improve the paper and help address your concerns. See the comment to all reviewers for details and results from the new experiment.
>
> **3.** We’d also make the point that Celligner (the non-variational baseline that we compared to for the TCGA-CCLE alignment experiment) uses mutual nearest neighbours (MNN), explicitly wrapping the function from the Seurat library, one of the standard methods used in the single-cell literature. For the new experiment of batch correction on scRNA-seq data, we have benchmarked CoMP against Harmony and Seurat, and our method compares highly favourably to both these methods across all our metrics. This result is consistent with the observed relative performance of CoMP vs. Celligner.
>
> **4.** Obtaining batch corrected data in the original gene expression space is useful. Some commonly used scRNA-seq batch correction / data integration methods do allow you to do this (Seurat), whilst others do not (Harmony).
> CoMP *does* allow this. Indeed, when performing batch correction with CoMP, one can simply decode the learnt representations, treating them as if they all came from a single batch, to obtain a batch corrected reconstruction $\hat{X}$ of the original data. This reconstruction can then be used for downstream tasks such as differential gene expression analysis, whilst the representations can be used for tasks such as pseudotime inference. Our final manuscript will also include the metrics evaluating batch correction on the original gene expression space for completeness.
>
> **5.** We agree that providing these plots would be very helpful. We have now generated the plots for a range of cancer types and we will be sure to include them in the appendix in the final version of the paper.
>
> **6.** We agree, we have generated the relevant UMAPs which show a high degree of clustering based on condition. We will include these for the final paper.
>
> **7.** Thanks for picking this up! This was a typo which we will correct for the final paper. The VFAE results are from published literature, the trVAE results are our own implementation.
>
> **References**
>
> [1] Leek, Jeffrey T., and John D. Storey. "Capturing heterogeneity in gene expression studies by surrogate variable analysis." PLoS genetics 3, no. 9 (2007): e161.

---

> > ### Comment · Reviewer_SVjF · 2021-08-23
> > **Thanks for your reply.**
> >
> > The authors addressed the concerns about previously not comparing CoMP with state of the art batch correction tools. They now included results of Harmony and Seurat, and it CoMP seems to be better in both kBET and silhouette score.
> > They also provide justification for dealing with possible over-correction.
> > They also promised to include plots for cancer types which were omitted from the original submission.
> > Based on the responses to my review and to other reviewers, I would like to modify the rating of the paper to 8.

---

### Official Review · Reviewer_vhhX · 2021-07-16

**Rating:** 6
**Confidence:** 4

**Summary:**

The paper proposes a penalty to learn representations that are
conditionally independent of some observed variables. The idea is to
use the contrastive loss that encourage the representations from the
same conditional variable to be spread out and those from different
conditional variable to be close. The paper demonstrates
application to fairness, counterfactual inference, and data
integration, and demonstrate the effectiveness of the penalty.


**Limitations And Societal Impact:**

The paper discussed the limitations in sec 7.

**Main Review:**

The paper proposes a penalty to learn representations that are
conditionally independent of some observed variables. The idea is to
use the contrastive loss that encourage the representations from the
same conditional variable to be spread out and those from different
conditional variable to be close. The paper demonstrates
application to fairness, counterfactual inference, and data
integration, and demonstrate the effectiveness of the penalty.

Both the theoretical and empirical results of the paper make sense. I
have a few major comments.

One comment is the choice of contrastive loss. It is understandable
that using the loss in Eq 3 would encourage tightly clustered
representations and produce suboptimally informative representations.
One thus may think contrastive loss may help avoid trivial solutions
and preserve informativeness. However, it is unclear why the second
penalty is sufficient to ``avoid tight clusters of points.'' Why is
the contrastive loss necessarily the right choice here? I think the
paper could benefit from a discussion about it.


A second comment is on the comparison between the misalignment penalty
and the other MMD penalties in the literature. The paper argues that
MMD suffers when there are many isolated cluster in the data. The
problem of MMD appears to be related to scaling; and the misalignment
penalty has the self-normalized property. I wonder if it implies that
MMD is equally effective if one manually rescale and recenter these
data points to achieve the normalization effect? What if the global
structure does not have many isolated clusters? Will the misalignment
penalty need to pay a price? Are there other differences between the
misalignment penalty and the MMD approaches that made the misalignment
penalty more preferable?


**Time Spent Reviewing:**

10

---

> ### Author Response · Authors · 2021-08-10
> **Individual reply to Reviewer vhhX**
>
> Thank you very much for a thoughtful review. We are really pleased to hear that both theoretical and empirical results of our paper make sense.
>
> > Why is the contrastive loss necessarily the right choice here?
>
> The first thing to emphasise is that the discussion in lines 161-192 provides an informal and intuitive justification for the CoMP penalty, whilst Theorem 2 and the proof of the theorem provide the rigorous justification. Specifically, in Theorem 2, we demonstrated that our contrastive penalty forms a stochastic upper bound on the weighted sum of KL divergences between posteriors from different classes- the specific form of our contrastive penalty was chosen to make sure that this theorem does hold, so minimising the CoMP penalty does lead to the desired properties in the latent space. On the point about “tight clusters of points”, we show in the proof of Theorem 2 that the term we add in the second penalty is a bound on the (neg) entropy of $q(z|c)$, so minimising the CoMP penalty can also be seen as ensuring that each $q(z|c)$ has high entropy, which is a more formal way of looking at the notion of “avoiding tight clusters of points”.
>
> > I wonder if it implies that MMD is equally effective if one manually rescale and recenter these data points to achieve the normalization effect?
>
> Theoretically, the MMD gradients and CoMP gradients would be different even in the case in which $z$ is normalised, because the denominator in (7) depends on $z_i$ itself. Put another way, the normalisation of the CoMP gradients applies separately to each $z_i$, rather than applying some normalisation across the whole latent space.
> Empirically, one way to investigate whether the difference between MMD and CoMP is due to latent space scaling is to vary the length-scale of the kernel used in the MMD penalty. We actually performed careful tuning of the kernel length-scale hyperparameter and the VFAE result presented in e.g. Table 2 is for the best performing value we found. This indicates that the performance of CoMP over MMD-based methods will persist, and possibly widen, with any additional global latent space scaling.
>
> > What if the global structure does not have many isolated clusters? Will the misalignment penalty need to pay a price?
>
> We would not expect that the CoMP method pays a price in the case of a simpler global structure. Indeed, in the income dataset, the latent space structure was simpler. In this experiment, CoMP outperformed the VFAE (see Figure 6 and Table 2). From a more formal perspective, Theorem 2 guarantees that, isolated clusters or not, the CoMP penalty will reduce the weighted sum of KL divergences, encouraging latent alignment.
>
> > Are there other differences between the misalignment penalty and the MMD approaches that made the misalignment penalty more preferable?
>
> Practically speaking, a key difference is that MMD requires an extraneous kernel to be specified and tuned (e.g. by tuning its length scale) whereas CoMP uses the variational posteriors themselves to define an alignment penalty.
> Beyond our intuition about global structure, the self-normalisation property in (7) may lead to more stable training.
> Mathematically speaking, MMD and CoMP are based on different divergence measures (MMD versus weighted sum of KLs)- for some comparison between MMD and KL in general https://arxiv.org/pdf/1406.2083.pdf could be helpful.

---

> > ### Comment · Reviewer_vhhX · 2021-08-31
> > **Thank you for your response**
> >
> > Thank you for your response to the reviews. My (positive) evaluation of the paper stays the same.

---

### Author Response · Authors · 2021-08-10
**Overall reply to all reviewers**

Thank you very much for your reviews. There was broad agreement that the paper was well-written (SVjF, UbUq), that the overall direction and topic of enquiry are important (SVjF, UbUq), and that the theoretical and empirical results make sense (vhhX). Reviewers appreciated the connection between counterfactual inference, batch correction and fairness that our paper makes explicit (SVjF, UbUq). The novel CoMP penalty was seen as “relevant and well-established” (SVjF), and the empirical results demonstrated the effectiveness of the method (vhhX).

A key strand of criticism focuses on the applicability of our method to single-cell RNA-seq batch correction, particularly in comparison with other non-VAE methods for this problem. To tackle this directly, we have run a **new scRNA-seq batch correction experiment** to benchmark CoMP against **Harmony** [1] and **Seurat** [2], two of the most frequently-used methods in this application domain. We performed our experiments on the dataset of peripheral blood mononuclear cells (PBMCs) processed with different technologies (10X library construction protocols). These data were provided by the authors of [1] and the application of Harmony to this dataset was presented in Fig. 4 of the Harmony paper [1]. We can report the following results:
* CoMP outperforms both Harmony and Seurat on the overall mixing metrics with the best (i.e. lowest) scores in both the kBET metric (CoMP: 0.171, Harmony: 0.318, Seurat: 0.436) and the silhouette score (CoMP: 0.0007, Harmony: 0.016, Seurat: 0.018).
* CoMP preserves biological information in the data by aligning samples from the same cell subtypes. Taking the mean over cell subtypes, CoMP outperforms both methods on the mean-kBET (CoMP: 0.132, Harmony: 0.245, Seurat: 0.356) and the mean-silhouette score (CoMP: 0.0029, Harmony: 0.0128, Seurat: 0.0221). This is an important metric, because it shows that CoMP correctly aligns matching cell types across different batches.
* CoMP preserves the biological cell subtype clusters from the original dataset. This can be visually inspected in the UMAP plots of the latent variables in the following anonymised links: (https://drive.google.com/drive/folders/1f1QG1NxqyTggkqWTz1oouNDqqKzP5Nwe?usp=sharing)

We have also provided a detailed response to the conceptual/mathematical side of this question within individual responses. To summarise, CoMP can be applied even when there is not exact independence between $z$ and $c$. We will update the paper with a detailed discussion on this point.

For other comments and questions, we have responded directly to individual reviewers.

[1] Korsunsky, Ilya, Nghia Millard, Jean Fan, Kamil Slowikowski, Fan Zhang, Kevin Wei, Yuriy Baglaenko, Michael Brenner, Po-ru Loh, and Soumya Raychaudhuri. "Fast, sensitive and accurate integration of single-cell data with Harmony." Nature methods 16, no. 12 (2019): 1289-1296.

[2] Stuart, Tim, Andrew Butler, Paul Hoffman, Christoph Hafemeister, Efthymia Papalexi, William M. Mauck III, Yuhan Hao, Marlon Stoeckius, Peter Smibert, and Rahul Satija. "Comprehensive integration of single-cell data." Cell 177, no. 7 (2019): 1888-1902.

---

### Decision · Program_Chairs · 2021-09-27

**Decision:**

Reject

**Comment:**

[This meta-review was written jointly by the SAC and AC after extensive discussion of the paper.]

This paper has interesting ideas and compelling results, and reviewers were overall supportive of the proposed work. Unfortunately, the paper has two entangled parts, where one of them has quite serious issues. The first part includes a Contrastive Mixture of Posteriors (CoMP) method that leverages a novel misalignment penalty to enforce certain independence relations; the second is related to counterfactual inference, data integration, and fairness.

Regarding the first part, CoMP enforces that a discrete auxiliary variable is independent of the latent variable inside a (conditional)-VAE without resorting to external discrepancies like an MMD. This is achieved by marginalizing out the inference network using empirical samples from the conditional distribution of the observation given settings of the discrete variable. This part of the paper is smooth, and the contributions are pretty substantial. It also connects quite nicely with the idea of variational inference and does not require an extraneous discrepancy measure or a separately tuned kernel, such as in previous approaches. The biggest benefit of this approach is that the scales of the terms in the loss are on the same scale as expectations of log probabilities or bounds of the same quantity. The approach could benefit many places where conditional-VAEs are used such as molecule generation (J. Lim, S. Ryu, J. W. Kim & W. Y. Kim, 2018) or dialog models (T. Zhao, R. Zhao & M. Eskenazi, 2017).

The second part of the paper regarding counterfactual inference, data integration, and fairness is quite concerning, and we feel there are fundamental misunderstandings regarding each one of them. The very claim that the paper introduces a unifying framework for solving these tasks is, from the beginning, completely off.

To understand why this is the case, consider the critical result used in the paper to compute Eq. 1, a quantity known as the "probability of causation," and written as $p(x_{c=c’} | x_i, c_i) = \int p(z|x_i, c_i) p(x | z, c’) dz$. By and large, the understanding of how to evaluate the l.h.s. of this expression is unsound. If one has the true, underlying structural causal model (SCM), say M*, the probability of causation could be evaluated in the manner described in Eq. 1, as introduced in Sec.7.1.1 of Causality (Pearl, 2000). On the other hand, whenever M* is unknown, this is not the case.  In particular, the probability distributions considered in the r.h.s. equation are arbitrary, representing a *learned* SCM M. The true SCM M* and the learned M are not necessarily the same.

The paper then proposes a method such that M will be trained to fit the data and guarantee the independence between Z and X. Still, it can be shown that it’s generically the case that there exists an M that will *almost always* disagree with the actual generating model M*, while agreeing with respect to the data. This means, therefore, that only with observational data, one cannot make statements about counterfactuals in generality. This is a well-known result in the field. For concreteness, please refer to the example discussed in Sec. 1.4.4 (Pearl, 2000) or examples 7-9 in (Bareinboim, Correa, Ibeling, Icard, 2020). This result is called the causal hierarchy theorem and can be found in Thm. 1 of the previous reference or Thm. 3 in https://arxiv.org/abs/2107.08558. Also, this has been shown in the context of autoencoders with continuous variables, for example, Appendix A.4 of https://arxiv.org/pdf/1907.03451.pdf,  has a construction where $c$ and $z$ are independent, but $x$ is constructed in a way that it is also marginally independent of $c$. This means the pair $(z, c)$ are independent and reconstruct $x$ and the pair $(x, c)$ are independent and reconstruct $x$. The pairs are observationally equivalent even with the extra independence constraint between the latent variable and $c$, but can be different under interventions.

Interestingly, the paper has one sentence (around line 124) acknowledging that it is not always possible to recover the correct model, while the rest of the paper seems to ignore this observation.  By and large, the challenge is that for almost any actual model M*, there are many models M that could fit the observed data equally well, but have very different implications in terms of their interventional and counterfactual distributions. This is certainly the most fundamental issue that disallows the paper to claim to be doing “counterfactual inference in generality.”

There are also serious issues regarding the tasks related to fairness and data fusion, and the purported “unification.” For example, many papers here at NeurIPS and other AI venues propose analyzing the fairness problem through causal lenses. The understanding from this work clarifies that enforcing independence among the protected attribute and the outcome variable is orthogonal to the issue at hand; in other, even when the independence is accomplished, biases may be amplified. Here are a few references that look at fairness from the viewpoint of causality (not an exhaustive list):

- Counterfactual Fairness, M. Kusner, J. Loftus, C. Russell, & R. Silva, NeurIPS’17.
- A Causal Framework for Discovering and Removing Direct and Indirect Discrimination. L. Zhang, Y. Wu, & X. Wu. IJCAI’ 2017
- Avoiding discrimination through causal reasoning. N. Kilbertus, M. Rojas-Carulla , G. Parascandolo, M. Hardt, & B. Scholkopf. NeurIPS’17.
- Equality of Opportunity in Classification: A Causal Approach. J. Zhang & E. Bareinboim, NeurIPS’18.
- Path-specific counterfactual fairness. S. Chiappa. AAAI’19.

Having said that, we note that it’s possible to stay oblivious to the causal inference literature and claim a purely statistical notion of fairness, which is anchored in conditional independence relations. In its current form, the paper claims to be doing counterfactual/causal inferences, so it’s expected the paper engages with the literature on fairness from the causal perspective, which goes beyond conditional independence relations and purely statistical measures.

Finally, the data integration problem has also been studied through the causal framework, and I will suggest Ch. 10 of the Book of Why (Pearl & Mackenzie, 2018) or (Bareinboim & Pearl, PNAS’16), which survey the challenges involved in such endeavors. Even though data integration is a different task, most of the issues with the proposed approach come precisely from the misinterpretation of Eq. 1, following the aforementioned discussion.

All in all, we would like to make it clear that the first dimension of the paper, including CoMP, is appreciated, and we believe the contributions are quite significant. On the other hand, the second part suffers from fundamental flaws. Therefore, we don’t know how to accept the paper without re-thinking and re-writing the paper in a major way. One thought would be to demonstrate the value of CoMP on tasks that are more bread-and-butter for conditional-VAEs like generation (see the dialog and molecule generation references above).